# MSLKNet: A Multi-Scale Large Kernel Convolutional Network for Radar Extrapolation

**Wei Tian** [1,2,*] **, Chunlin Wang** [1,2] **, Kailing Shen** [1,2] **, Lixia Zhang** [3] **and Kenny Thiam Choy Lim Kam Sian** [4]

1    School of Software, Nanjing University of Information Science and Technology, Nanjing 210044, China; chunlin@nuist.edu.cn (C.W.); skling@nuist.edu.cn (K.S.)
2    Engineering Research Center of Digital Forensics, Ministry of Education, Nanjing University of Information Science and Technology, Nanjing 210044, China
3    Shijiazhuang Meteorological Bureau, Shijiazhuang 050081, China; 13933119682@163.com
4    School of Atmospheric Science and Remote Sensing, Wuxi University, Wuxi 214105, China; kennlks@gmail.com
*    Correspondence: tw@nuist.edu.cn

**Abstract:** Radar echo extrapolation provides important information for precipitation nowcasting. Existing mainstream radar echo extrapolation methods are based on the Single-Input-Single-Output (SISO) architecture. These approaches of recursively predicting the predictive echo image with the current echo image as input often results in error accumulation, leading to severe performance degradation. In addition, the echo motion variations are extremely complex. Different regions of strong or weak echoes should receive different degrees of attention. Previous methods have not been specifically designed for this aspect. This paper proposes a new radar echo extrapolation network based entirely on a convolutional neural network (CNN). The network uses a Multi-Input-Multi-Output (MIMO) architecture to mitigate cumulative errors. It incorporates a multi-scale, large kernel convolutional attention module that enhances the extraction of both local and global information. This design results in improved performance while significantly reducing training costs. Experiments on dual-polarization radar echo datasets from Shijiazhuang and Nanjing show that the proposed fully CNN-based model can achieve better performance while reducing computational cost.

**Keywords:** radar echo extrapolation; multi-scale large kernel convolution; long-term prediction





## 1. Introduction

Precipitation nowcasting, which refers to high-resolution weather forecasting within a short period of 0–2 h [1], is one of the most important tasks in weather forecasting, and has received increasing attention in the research community [2]. The significance of precipitation nowcasting lies in its ability to provide accurate advance predictions of short-term precipitation within a forecasted period, which is crucial for addressing meteorological disasters, optimizing resource management, improving transportation logistics, and enhancing public safety [3–5]. This forecasting technology aids various sectors in effectively dealing with unpredictable weather conditions, reducing potential risks and losses, and boosting overall productivity.

Weather radar has become a major tool for precipitation nowcasting because of the high temporal and spatial resolution of its echo images. Radar echo refers to the radar phenomenon where electromagnetic waves emitted into the atmosphere encounter precipitation particles, such as raindrops or snowflakes. The waves scatter, reflect, or are absorbed by these particles, returning to the radar and forming an image known as an echo image. This image is crucial for displaying and analyzing precipitation location, intensity, and distribution in meteorology and radar meteorology. Echo position refers to the specific location of precipitation echoes displayed on an echo image. The echoes and their positions

are crucial for displaying and analyzing the location, intensity, and distribution of precipitation. Traditional radar echo extrapolation methods mainly include cross-correlation algorithms [6], cell centroid tracking algorithms [7], and optical flow algorithms [8], which extrapolate the echo position of the next moment based on the radar echo images of several moments. Despite the high computational efficiency of these methods, they ignore the complex nonlinear variations in the small- and medium-scale atmospheric systems in the radar echo. They also suffer from the underutilization of historical radar data and the limitation of short extrapolation time limit [9].

In recent years, with the rapid development of artificial intelligence technology, deep learning methods have been heavily applied to various fields, including precipitation nowcasting. Most of the current deep learning methods focus on extrapolation tasks guided by radar echo data because of the ease of collecting a large amount of continuous radar echo data to meet the data requirements of long-time forecasting tasks.

The main models currently used for radar echo extrapolation tasks use a hybrid architecture of convolutional neural networks (CNN) and recurrent neural networks (RNN). Such an architectural design allows the models to exploit both the ability of convolutional units to model spatial relationships and the potential of recursive units to capture temporal dependencies. Although such predictive architectures give satisfactory results, they are still limited and not fully suitable for radar echo extrapolation tasks [10].

Firstly, these recurrent models are based on the Single-Input Single-Output (SISO) architecture, which generates the next frame using the current prediction frame by learning the hidden state of the historical information. However, as more frames are generated, their quality and accuracy deteriorate rapidly due to the effect of small errors in the earlier frames. In particular, the complexity of the radar echo motions is such that even small errors can easily be amplified into severe compound errors over time. To effectively suppress cumulative errors, some methods for prediction tasks such as Simvp [11] and MIMO-VP [12] adopt a Multi-Input Multi-Output (MIMO) approach to model building. These models have shown significant performance gains. They encode the spatiotemporal representation by stacking feature maps of all input frames in the translator module and simultaneously decoding them into multiple future frames. However, only a few attempts in video prediction have adopted the MIMO architecture, and there has been limited exploration of the MIMO architecture in the radar extrapolation field.

Secondly, previous models have explored the hidden representation of spatiotemporal variations, ignoring the extraction of global features. Convolution kernel size is a very important design dimension, but is often neglected. The traditional convolution-learning representation used in the model is strongly biased against local features. At the same time, this leads to severe damage to global features [13]. Larger convolutional kernels cover a wider spatial region of the image with a larger receptive field. Thus, more contextual information can be taken into account, including features and structures at larger scales. This facilitates the model to understand the overall appearance and overall nature of the overall image, rather than just focusing on local details. Due to weather systems and complex terrain factors, the shape, size, and motion speed of radar echoes vary greatly from one rainfall process to another. The perceptual field provided by traditional small convolution alone is very limited, and the ability to capture the spatial and temporal variation in radar echoes when they move over a large area cannot meet the operational requirements. Therefore, the radar echo extrapolation task needs to consider more spatial feature information and different levels of attention should be paid to the strong and weak echoes in different regions.

In this paper, we propose a new convolution-based spatiotemporal feature extraction network, MSLKNet, to fully exploit the spatiotemporal relationships of meteorological data and reduce prediction errors over time. We use a Multi-Input-Multi-Output (MIMO) approach to build our model, which updates the design of traditional convolution in previous radar echo extrapolation models and differs from stacked RNN models using traditional small convolution to extract features. A multi-scale large kernel convolution attention

module is used to evoke spatial attention to preserve the global spatial information of the frame. Building multi-scale spatial attention is more effective than standard convolution and self-attention in spatial information encoding. In addition, an information recalling scheme is applied to facilitate prediction. The contributions of our work can be summarized as follows:

- We reconsider the convolutional attention structure and design the multi-scale large kernel (MSLK) convolution module to acquire a multi-scale radar echo background from local to global.
- We propose a new CNN-based radar extrapolation architecture to reduce extrapolation accumulated errors by building a MIMO-based model. Moreover, an information recalling scheme is applied to further preserve the visual details of the predictions.
- Comprehensive experiments are conducted on two real dual-polarization radar echo datasets.

## 2. Related Work

In recent years, machine learning and deep learning approaches have become dominant in the field of weather prediction and meteorological research. This trend is due to the large amount of accessible radar or satellite image data and the rise of advanced models such as deep neural networks, which provide powerful tools for solving diverse problems in this field.

There are two main structures of existing deep learning models for radar echo extrapolation, the CNN based on the UNet [14] structure and the stacked ConvRNN model.

UNet and its variants are prominent CNN models that focus on utilizing convolutional modules to learn temporal and spatial data variations. These models are renowned for their outstanding performance and widespread application in various domains. Through convolutional modules, they efficiently capture data features and patterns, enabling the analysis and prediction of temporal and spatial data. SE-ResUNet [15] embeds ResNet (Residual Network) [16] modules into U-Net to improve prediction accuracy. SmaAt-UNet [17] equips UNet with an attention module and depth-separable convolution, using only a quarter of the training parameters without compromising. FureNet [18] adds two additional encoders to UNet for multimodal learning. WF-UNet [19] uses a 3D Unet model to integrate precipitation and wind speed variables as inputs to the learning process and analyzes the impact on the precipitation target task. Broad-UNet[20] is equipped with asymmetric parallel convolution as well as the Atrous Spatial Pyramid Pooling (ASPP) [21] module, which learns more complex patterns by combining multi-scale features while using fewer parameters than the core UNet model. These models are simple in structure and easy to apply. However, convolution is more concerned with extracting spatial features and has a natural drawback in capturing temporal trends; this is not fully applicable to time series tasks.

The stacked ConvRNN model attempts to design a new spatiotemporal module and then stack multiple such modules to form the final model. ConvLSTM [22] is the pioneer of this work, which pioneered the use of convolution and LSTM to model spatial and temporal variations, respectively. TrajGRU [23] absorbs the advantages of the trajectory tracking strategy by substituting optical flow into the hidden state. PredRNN [24] proposed a spatiotemporal memory unit that propagates information horizontally and vertically through highway connections that can extract and store both spatial and temporal representations, and its follow-up work PredRNN++ [25] further proposed a gradient highway unit and Casual LSTM to capture temporal dependence adaptively. MIM [26] introduced more memory cells to handle smooth and non-smooth information to enhance the ability of PredRNN to model higher-order dynamics. E3D-LSTM [27] designed eidetic memory transformations to further enhance the long-time memory capability of LSTM and proposed 3D convolution to enhance its performance. MotionRNN [28] modeled overall motion trends and transient changes uniformly. PrecipLSTM [10] designed two modules focusing on meteorological spatial relationships and meteorological temporal variations, respectively, and combined

the two modules with PredRNN to adequately capture the spatiotemporal dependence of radar data. The ConvRNN model can effectively capture the spatio-temporal features in the input data and is suitable for tasks with spatial and temporal dependencies. However, its high computational cost in dealing with long sequences or large input data will lead to a decrease in the efficiency of training and inference [29].

To summarize, previous methods have certain limitations. In the past, to capture long-term spatio-temporal dependencies, CNNs typically introduced attention mechanisms. These approaches have had limited results, and we used large kernel convolution to build multi-scale spatial attention when encoding spatiotemporal information. Subsequently, we learned the stacked ConvRNN model, which enables the modeling of spatio-temporal dependencies by stacking such specially designed convolutional modules.

## 3. Approach

### 3.1. Problem Formulation

Precipitation nowcasting describes the present weather conditions and the weather forecast from 0 to 2 h. Therefore, the radar extrapolation problem can be defined as using the radar echo reflectivity of 1 h before the current time to predict the reflectivity of the following 2 h. A radar extrapolation model typically takes a video clip $\{v_1, \ldots, v_i\}$ as the inputs and outputs the future video clip $\{\hat{v}_{i+1}, \ldots, \hat{v}_T\}$. The problem we want to optimize can be represented by Equation (1):

$$\min \sum_{t=i+1}^{T} [\mathcal{L}(\hat{v}_t, v_t)], \tag{1}$$

where $\hat{v}_t$ denotes the predicted frame at time step $t$, $\mathcal{L}$ denotes the loss function, such as the $\mathcal{L}_1, \mathcal{L}_2$ loss functions and so on.

### 3.2. Overview

We illustrate the detailed architecture of MSLKNet in Figure 1, where the input past frames are first encoded into a low-dimensional potential space by an encoder consisting of three convolutional layers. The spatial features of the radar echo maps are extracted at each convolution step while reducing the spatial resolution of the feature maps. Next, the translator captures spatial dependencies and local motion variations by learning the latent space. The translator consists of several specially designed stacks of multi-scale large kernel convolution blocks. The MSLK Block is divided into two parts to extract global structural features and local motion variations, described in more detail in the following two sections. Finally, the decoder decodes the potential space into predicted future frames. The decoder is symmetric with the encoder, including three deconvolution layers. An information recall scheme is used to bridge the gap between low-level details and high-level semantics, preserving spatial features. Figure 1(2) depicts the detailed structure of the MSLK block. The first part of MSLK focuses on the global spatial structure and uses operations such as multi-scale convolution to capture the spatial information of the image. The second part of local motion concern (LMC) focuses on local motion changes and uses convolution to extract features at each location, increasing the nonlinear representation of the model and improving the diversity of features by up- and down-dimensioning. Figure 1(3) depicts the detailed structure of MSLK.

### 3.3. Multi-Scale Large Kernel Convolution (MSLK)

Unlike many video prediction tasks, the radar echo extrapolation task needs to focus on the local echo motion and the global system motion trend due to the complexity of atmospheric system evolution, which cannot be satisfied by the traditional representation of convolutional learning. The attention mechanism [30] may be more suitable for global feature extraction. It is a hot topic in visual transformer research, and aims to make the network focus on the important parts, capturing the influence of all other points across the map and adaptively selecting differentiated features based on the input features.

Unfortunately, the attention mechanism requires an excessive memory footprint, and models with stacked RNN units cannot meet this demand. Recent studies have found that large convolutional kernel models, supported by novel model designs, demonstrate comparable or superior performance to Transformer models in various deep learning tasks. Large kernel convolution offers significant advantages in obtaining larger effective sensory fields and more accurate shape biases [31–35]. Therefore, we consider using large kernel convolution to model attentional mechanisms to achieve larger receptive fields and capture multi-scale spatial structures.

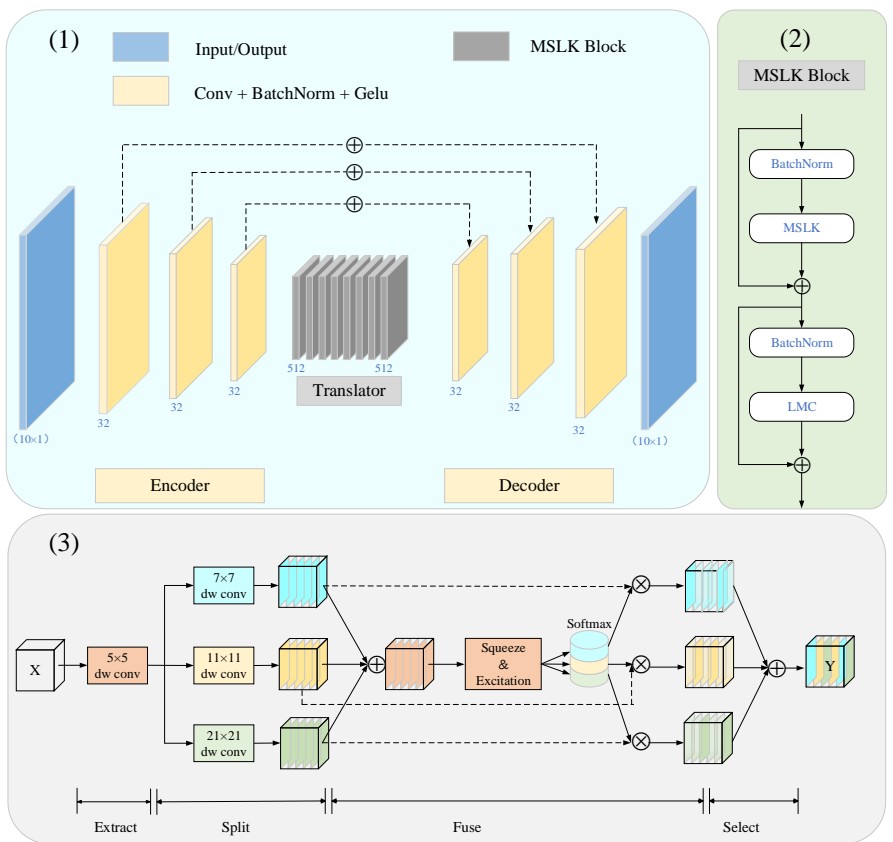

**Figure 1.** Network structure of our model. (**1**) The consecutive input frames are first sent to the encoder for encoding. After encoding, the translator extracts spatial features and spatio-temporal correlations. Finally, the features are decoded into predicted video frames. The encoder and decoder consist of three convolutional layers. The translator consists of stacked MSLK Blocks. (**2**) Architecture of MSLK Block. (**3**) Architecture of MSLK in (**2**).

To enable the model to obtain a larger receptive field and to be able to adjust its receptive field size adaptively, we propose a multi-scale convolution with automatic selection [36] among multiple kernels with different kernel sizes. The specific structure is shown in Figure 1. *MSLK* consists of four parts: extract, split, fuse, and select. First, a deep convolution with a convolution kernel of 5 is used to aggregate local information, followed by the application of a deep strip convolution with multiple branches for capturing multi-scale spatial information. We use a fusion operation to aggregate information from multiple paths to obtain a global and integrated representation and select weights. Finally, we aggregate feature maps of different-size kernels based on these weights. This allows for better integration and utilization of information from different sensory fields to improve the performance and robustness of the model. *MSLK* can be described by Equation (2):

$$y = F_{MSLK}(x) = \sum_{i=1}^{3} \frac{\alpha_i}{\sum_{j=1}^{3} \alpha_j} f_i(\text{DW-Conv}(x)), \tag{2}$$

where $x$ denotes the input feature map. DW-Conv denotes the depth direction convolution and $f_i, i \in \{1, 2, 3\}$ denotes the $i$-th branch. In each branch, we use two depth-wise strip convolutions instead of the conventional convolution, where the convolution kernel size of each branch is set to 7, 11, and 21, respectively. Depth strip convolution can significantly reduce the convolution parameters and significantly improve efficiency with limited loss of model performance. $\alpha_i, i \in \{1, 2, 3\}$ denotes the weight of the $i$-th branch.

### 3.4. Local Motion Concern (LMC)

Another important aspect of radar echo extrapolation is capturing short-term local motion changes, which is critical for generating future video frames. That is, the model needs to have the sensitivity and ability to capture local motion changes and effectively identify and predict local motion changes in radar echoes to generate video sequences with continuity and naturalness. Therefore, it is justified to install a local motion concern block behind the MSLK block to effectively extract local features in the image.

In designing the block, we adopt the idea of Fully Connected Feedforward Network (FFN) in the Transformer [37] structure. Specifically, we first map the low-dimensional features into the high-dimensional space by $1 \times 1$ convolution to increase the complexity and richness among the features. Next, we use $3 \times 3$ deep convolution to extract local information and perform nonlinear transformations on the features by Gelu functions to improve the expressiveness of the network. Finally, we downscale the features by $1 \times 1$ convolution to remove the less relevant information and focus more on the important information of local motion. In this process, we use the residual connection to combine contextual information to further improve the effectiveness and accuracy of the model.

### 3.5. Information Recall Scheme

Considering the problem of information loss during encoding, we adopt an information recall scheme between the encoder and decoder and can be represented by Equation (3):

$$D_l = \text{Dec}(D_{l-1} + E_{-l}), \quad l = 1, \ldots, \tag{3}$$

where $D_l, E_{-l}$ denote decoded features from the $l$th layer of the decoder and the encoded features from the $l$th from the last layer of the encoder. $Dec_l$ denotes the $l$th layer of the decoder. On the basis of the above information recalling scheme, the gap between low-level detail and high-level semantics is bridged. The decoder can recall multi-level encoded information back and improving the quality of forecasts.

## 4. Experiment

In this section, we validate the performance of our proposed model on two real radar datasets. This study uses data from a C-band dual polarization weather radar operated by Nanjing University and a S-band dual polarization weather radar operated by Shijiazhuang Meteorological Bureau. Currently, S-band and C-band radars are the main weather radars in operation in China.

### 4.1. Datasets

#### 4.1.1. Nanjing Dual-Polarization Radar Dataset

NJU-CPOL [18] is a C-band dual polarimetric weather radar open dataset provided by Nanjing University. The dataset contains 268 precipitation events from 2014 to 2019, using Constant Altitude Plan Position Indicator (CAPPI) data at 3 km altitude with prior quality control [38] of the raw data and interpolation into a Cartesian coordinate system. CAPPI displays radar echo information at a particular altitude, which helps to observe weather conditions at a particular altitude and can be used to analyze precipitation, thunderstorm activity, cloud structure, etc. The temporal resolution of the dataset is 6–7 min, the spatial resolution is 1 km horizontally, and the area around the radar center is $256 \times 256$ km.

Many precipitation events in the dataset have missing data and the event sequence length cannot meet the requirements for the proximity forecasting task. Therefore, after the

screening, we divide the dataset into a training set containing 5547 sequences and a test set containing 1110 sequences.

### 4.1.2. Shijiazhuang Dual-Polarization Radar Dataset

The dataset generated from S-band dual-polarization radar-based data is provided by the Shijiazhuang Meteorological Bureau, Hebei Province, China. We collected radar data for all precipitation days from 2020 to 2022 and took the maximum of the reflectances from the nine elevation angles to form a combined reflectance to be applied for radar echo extrapolation. The raw radar data were first quality-controlled, and non-meteorological echoes in the radar echoes were removed using dual-polarization radar parameters such as Differential Reflectivity (ZDR), Specific Differential Phase (KDP), and correlation coefficients. The data were rasterized into a $200 \times 200$ grid using the K-nearest neighbor regression algorithm, which covered the entire Shijiazhuang city (113.5° E–115.5° E and 37° N–39° N, with a resolution of 0.1°). The temporal resolution of the radar-based data are 6 min. For the partitioning of the dataset, we refer to the method in ConvLSTM [22], which divides each precipitation day sequence into six blocks, randomly assigning five blocks for training and one block for testing. Then, we slice the consecutive frames in each block with a 20-frame wide sliding window to generate a training set of 5254 samples and a test set of 814 samples.

### 4.2. Experiment Setup

We first refer to the general setup of a spatiotemporal sequence prediction task on a radar dataset, generating 10 future frames by inputting 10 prior frames when training the model. We then extend the extrapolation length from 10 to 20 frames to explore the model's ability in long-term prediction covering the next 2 h.

We compare MSLKNet with five benchmark models in the literature: ConvLSTM, PredRNN, PredRNN++, MIM, and MotionRNN. We set the mini-batch to 8 to optimize the model using $\mathcal{L}_2$ loss as the training loss using the Adam optimizer [39] with a learning rate of 0.001. All experiments were performed on an NVIDIA 3060 GPU. We computed several evaluation metrics for the prediction results, such as Mean Square Error (MSE), Mean Absolute Error (MAE), Structural Similarity (SSIM), Peak Signal-to-Noise Ratio (PSNR) and Critical Success Index (CSI). These metrics are computed and averaged for all prediction frames to comprehensively assess the performance and effectiveness of the model. Among them, MSE and MAE reflect the prediction accuracy and precision of the model. PSNR is a metric for measuring image quality, commonly used to compare the similarity between an original image and a processed or compressed version. SSIM is another metric used for image quality assessment, considering not only brightness, but also contrast and structure. PSNR primarily focuses on brightness, being less sensitive to changes in contrast and structure. SSIM considers structural information, making it more perceptually aligned with human vision. CSI is a metric commonly employed in the assessment of the consistency between predictions and observations, particularly in the context of precipitation forecasting. In addition, we use the fvcore [40] library to report the number of triggers per sample to ensure the accuracy of the calculation. Also, we report the training time by calculating the average time (in seconds) required to train an epoch. By considering these metrics, we can evaluate the performance and effectiveness of the model more comprehensively.

### 4.3. Results

Tables 1 and 2 present the performance of each model on the two radar echo datasets from Shijiazhuang and Nanjing. MSLKNet achieves consistent improvement for all the metrics, indicating that our proposed model obtains the best prediction quality with the fastest training time and the smallest computational resource consumption.

Compared with MotionRNN, which has the best extrapolation effect, MSLKNet improves the MSE by 11.6%, and the SSIM increases from 0.836 to 0.857 on the Shijiazhuang

Radar dataset using only 1/3 of the computing resources and less training time. In addition, the overall nowcasting performance on the testing data set is quantitatively evaluated with CSI for 30 dBZ threshold. The CSI (R > 30) of MSLKNet increased from 0.051 to 0.058, which indicates that our model has a good nowcasting ability for heavier rainfall. Our model can predict future frames more accurately than other models while making better use of computational resources and training time. Our model does not use RNN, LSTM or complex modules and relies on CNN with good computational optimization, and avoids iterative computation, making the training process faster than other methods while saving computational costs.

**Table 1.** Quantitative results of the different models on the Shijiazhuang radar dataset (10→20 frames).

| Model | Flops (G) | Training Time (s) | MSE ↓ | MAE ↓ | SSIM ↑ | PSNR ↑ | CSI ↑ |
|---|---|---|---|---|---|---|---|
| ConvLSTM | 14.9 | 416 | 149.89 | 1300.66 | 0.751 | 32.84 | 0.021 |
| PredRNN | 30.1 | 508 | 127.19 | 1119.07 | 0.808 | 33.55 | 0.036 |
| PredRNN++ | 41.3 | 541 | 122.99 | 1095.49 | 0.807 | 33.70 | 0.046 |
| MIM | 44.9 | 587 | 102.80 | 955.57 | 0.833 | 34.01 | 0.042 |
| MotionRNN | 33.4 | 569 | 99.62 | 964.65 | 0.836 | 34.04 | 0.051 |
| MSLKNet | 12.7 | 371 | 88.11 | 864.83 | 0.857 | 34.51 | 0.058 |

Figure 2 shows the MSE, MAE, SSIM and PSNR of the next 20 frames for each model. Based on these metrics, it can be seen that over time, our model produces prediction results with a lower mean difference and variance loss as well as higher image quality, showing a clear advantage of the model in capturing long-term motion trends.

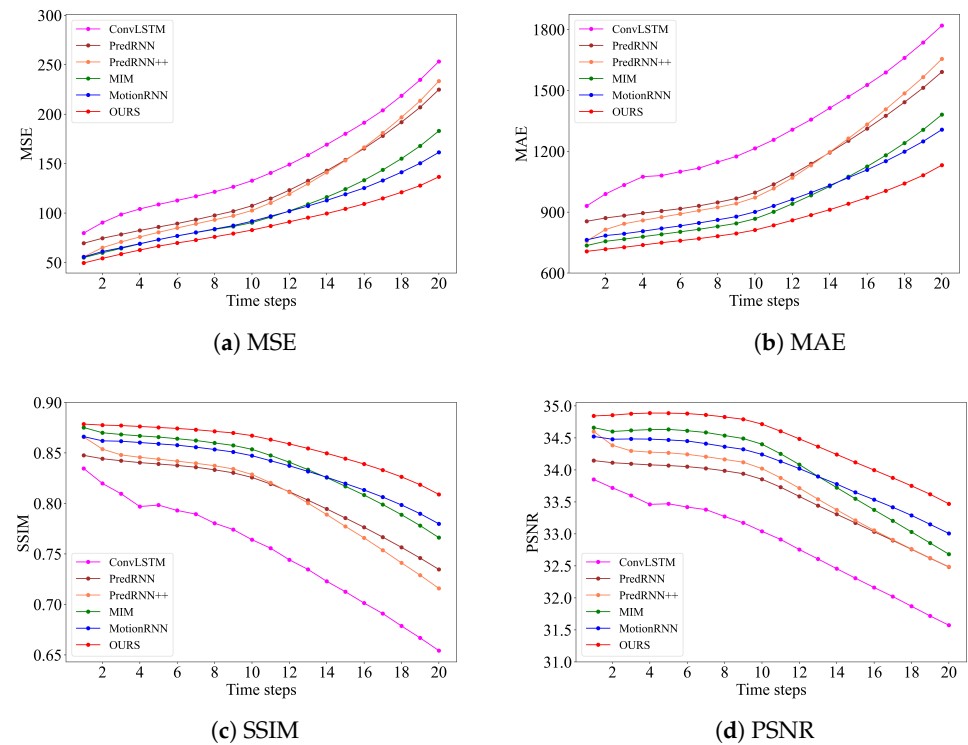

(**a**) MSE  (**b**) MAE

(**c**) SSIM  (**d**) PSNR

**Figure 2.** Frame-wise comparisons of the next 20 extrapolation echoes.

Figures 3 and 4 provide representative examples from the two test datasets. The quantitative visualization results show that the MSLKNet-predicted frames are resistant to ambiguity and can retain more overall structure and detailed features. For example, the prediction results for the development of the echo center region from the right part of the last predicted frame in Figure 3 are very similar to the true value, while other



models produce significantly blurred frames or have errors in the prediction of the echo center location.

**Table 2.** Quantitative results of the different models on the Shijiazhuang radar dataset (10→20 frames).

| Model | Flops (G) | Training Time (s) | MSE ↓ | MAE ↓ | SSIM ↑ | PSNR ↑ | CSI ↑ |
|---|---|---|---|---|---|---|---|
| ConvLSTM | 24.4 | 737 | 238.60 | 2338.89 | 0.772 | 34.86 | 0.083 |
| PredRNN | 49.3 | 911 | 190.26 | 2018.65 | 0.791 | 34.95 | 0.162 |
| PredRNN++ | 67.6 | 1401 | 166.52 | 1875.35 | 0.805 | 35.18 | 0.153 |
| MIM | 73.6 | 1432 | 153.98 | 1790.89 | 0.812 | 35.30 | 0.167 |
| MotionRNN | 54.7 | 1317 | 134.31 | 1704.20 | 0.821 | 35.34 | 0.181 |
| MSLKNet | 20.9 | 680 | 124.59 | 1653.01 | 0.825 | 35.56 | 0.192 |

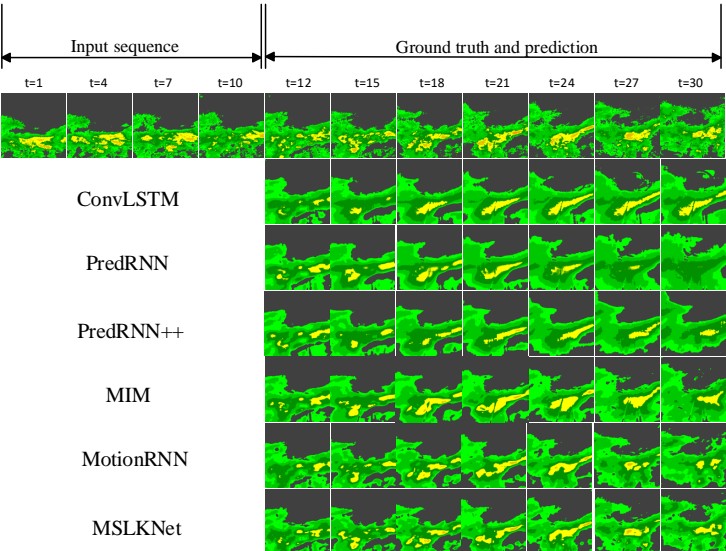

**Figure 3.** Visualization samples on the Shijiazhuang radar echo dataset. Yellow indicates higher echo intensities.

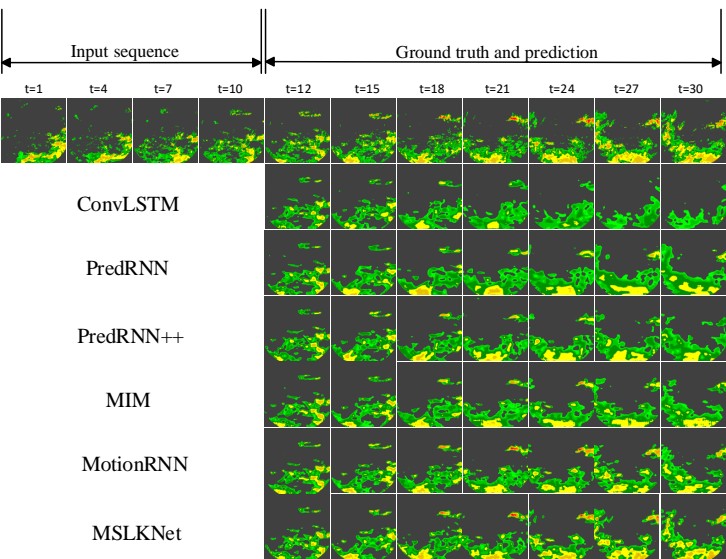

**Figure 4.** Visualization samples on the Nanjing radar echo dataset. Yellow indicates higher echo intensities.

### 4.4. Ablation Experiments

To verify the effectiveness of each module in MSLKNet, we conduct a comparison experiment on the Shijiazhuang dual-polarization radar echo dataset. From the quantitative

results Table 3, we find that our improvements can reinforce each other and the model can achieve better results in each metric.

**Table 3.** Ablation experiments for information recall scheme, multi-scale large kernel convolution module (MSLK) and local motion concern (LMC) on the Shijiazhuang radar dataset. We use $3 \times 3$ ordinary convolution to replace the MSLK Block in the basenet).

| Model | MSE ↓ | SSIM ↑ | PSNR ↑ |
|---|---|---|---|
| Basenet | 131.12 | 0.783 | 33.39 |
| MSLKNet w/o MSLK | 97.25 | 0.835 | 33.96 |
| MSLKNet w/o LMC | 106.01 | 0.823 | 33.82 |
| MSLKNet w/o recall | 92.07 | 0.854 | 34.46 |
| MSLKNet | 88.11 | 0.857 | 34.51 |

## 5. Conclusions

This paper proposes the MSLKNet, a convolution-based spatiotemporal feature extraction network for radar extrapolation. The network includes two modules, MSLK and LMC, which focus on multi-scale global spatial information extraction of radar echoes and local motion objects, respectively. In addition, we design the model using a MIMO approach to improve long-term prediction capability. Radar echo extrapolation experiments demonstrate the effectiveness of our approach. Our approach can predict future frames more accurately while using computational resources and training time more efficiently. In addition, the approach improves long-term prediction.

In the future, multiple parameters related to precipitation nowcasting in dual-polarization radar will be introduced, and further improvement of the model architecture will be studied to promote the fusion of multiple variable information.

**Author Contributions:** Conceptualization, C.W., W.T. and K.S.; methodology, C.W., W.T. and K.S.; validation, C.W., W.T. and K.S.; writing—original draft preparation, C.W., W.T. and K.S.; writing—review and editing, C.W., W.T. and K.S.; visualization, C.W., W.T. and K.S.; supervision, W.T., L.Z. and K.T.C.L.K.S.; project administration, W.T., L.Z. and K.T.C.L.K.S.; funding acquisition, W.T. and L.Z. All authors have read and agreed to the published version of the manuscript.

**Funding:** This work was supported in part by the National Key Research and Development Program of China under Grant 2021YFE0116900,in part by the National Natural Science Foundation of China under Grant 42375147, in part by the National Natural Science Foundation of China under Grant 42075138, and in part by the National Natural Science Foundation of China under Grant 42175157.

**Data Availability Statement:** The NJU-CPOL dataset is available at https://doi.org/10.5281/zenodo.5109403 (accessed on 13 October 2021). Shijiazhuang Dual-Polarization Radar Dataset is available on request from the corresponding author. The data are not publicly available due to the confidentiality policy of Shijiazhuang Meteorological Bureau.

**Conflicts of Interest:** The authors declare no conflicts of interest.

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
