# Peer review of "MSLKNet: A Multi-Scale Large Kernel Convolutional Network for Radar Extrapolation"

_atmosphere, doi:10.3390/atmos15010052_

Round 1
Reviewer 1 Report
Comments and Suggestions for Authors
Please note the following suggestions:
Line 25: Explain proximity forecasting. If it is the same as precipitation forecasting or an umbrella term for a general type of forecasting, please state it clearly.
Line 28: Unless your audience is well-versed with radar technology, explain echo and echo position briefly in your Introduction.
Line 44-48: Could you cite a quantifiable example of propagation of error over time or the cumulative impact of small errors over time?
Line 51: Please elaborate on what global features entail.
Line 53,54: Please elaborate what global and local textures mean. If the impact on local and global textures stems from the convolution kernel size, please clarify how. If not, please consider rephrasing the leading sentences for less ambiguity.
Line 61 - 70: Please explain the MIMO aspect and compare it to SISO. How is your work using 'multi' input/output as compared to others 'single' input/output. Please also clarify if this is the only work of its kind that uses a MIMO approach. If your approach has never been done before, you should state it. If it has been done before but not in an atmospheric context, you should state that as well. Overall, help your audience gauge the novelty of your results.
Line124 - 127: What are the limitations of convRNN? If they aren't clearly listed, please list them. If there are no big limitations, please justify how your model is superior or compares with convRNN.
Line 215: Please explain what a dual-polarization dataset is. Also explain the difference between an S and C band dual polarization dataset.
Line 253 - 262: Please explain the metrics, SSIM and PSNR, better. So far, the definitions are very high-level.
Lines 221: What does 'prior' quality control entail? Please clarify.
In the Results Section:
Please discuss why your model's training time and computational resources are lesser than other models you compare with. Please also discuss the characteristics of your model that make it perform better, compared to the other models. In addition to seeing the empirical evidence, your audience must also be able to understand why the empirical evidence shows what it does.
The results in Fig. 4 don't seem all that different for MSLKNet in comparison to the other models. Meanwhile, a larger contrast can be observed in Figure 3. What is that? If this is an artifact of the datasets used for each test, please state how the datasets impact the results.
Conclusion: Your conclusion is insufficiently detailed. Please include a summary of your experiments and results in this approach, highlighting in a little more detail the characteristics of your model that impact its overall performance.
Comments on the Quality of English LanguageThe paper is written well, overall. However, please note the following minor items needing attention / consideration:
Line 3-4 : I recommend using better descriptive terminology in place of next time and current time with reference to predictive and current echo images.
Line 8 - 11: Grammatically incorrect sentence.
Line 92: Please expand the abbreviation ResNet.
Line 121 - 123: Please reword this sentence. Your usage of the phrase 'this capability' isn't clear on which capability you are referring to.
Line 221, 235: Please expand the abbreviations CAPPI, ZDR, and KDP.
Reviewer 2 Report
Comments and Suggestions for Authors
In my opinion, the article has good quality and innovation, and the results obtained in the paper are very important and can be used by other researchers.
Comments on the Quality of English Language-
Author Response
Thank you very much for your valuable feedback and thorough review. I am honored by your suggestion to proceed with publication without further revisions. Your recognition of our research is greatly appreciated.
Reviewer 3 Report
Comments and Suggestions for Authors
GENERAL REMARKS
The manuscript deals with precipitation nowcasting based on weather radar data.
The limit between long-range nowcasting and forecasting might be considered. In forecasting the methods are mostly based on atmospheric models, and combining these with radar observations should not be overlooked. Especially the atmospheric dynamics forms the backbone of precipitation, and "nonlinear variations" in radar echo fields that simple extrapolation methods miss, should be included in the atmospheric dynamics, not just as appearing nonlinear relations to earlier radar echo fields.
All the important possible uses of nowcasting are mentioned, but you are not much looking if the datasets you select include the important cases, or how well these would be present in the evaluation of the models. Especially longer time span in nowcasting may not be easy to achieve in rapidly developing systems and short duration but heavy rain episodes.
COMMENTS ON DETAILS
4. Experiment.
a. Radar datasets. "CAPPI" might be explained, one might assume that it is something else than radar reflectivity related. On the other hand the other radar data seems to be extracted differently, but what you mean by "combined reflectance" is not completely clear.
b. Your evaluation metrics may be more like comparing the similarity of the image patterns. You might discuss if these have relations to metrics used in precipitation nowcasting/forecasting evaluation?
c. The use of multiple radar quantities is mentioned. This is an interesting development as the dual-polarimetric quantities are generally thought as telling more about the type of hydrometeors and also making the precipitation intensity detection more reliable. Already discussed in "general remarks" you should not forget the option of using atmospheric dynamics model, usually called "numerical weather prediction models", as the more accurate and versatile radar data in their input may have much improving impact in the model predictions.
